# Recent Non-Invasive Parameters to Identify Subjects at High Risk of Sudden Cardiac Death

**DOI:** 10.3390/jcm11061519

**Published:** 2022-03-10

**Authors:** Maria Delia Corbo, Enrica Vitale, Maurizio Pesolo, Grazia Casavecchia, Matteo Gravina, Pierluigi Pellegrino, Natale Daniele Brunetti, Massimo Iacoviello

**Affiliations:** 1Cardiology Unit, Department of Medical and Surgical Sciences, University Polyclinic Hospital of Foggia, University of Foggia, 71100 Foggia, Italy; mariadeliacorbo@gmail.com (M.D.C.); enri91@gmail.com (E.V.); m.pesolo@gmail.com (M.P.); graziacasavecchia@libero.it (G.C.); pierluigi.pellegrino@yahoo.it (P.P.); natale.brunetti@unifg.it (N.D.B.); 2University Radiology Unit, University Polyclinic Hospital of Foggia, 71100 Foggia, Italy; matteogravina@inwind.it

**Keywords:** sudden cardiac death, prognosis, ECG, cardiac magnetic resonance, genetic testing

## Abstract

Cardiovascular diseases remain among the leading causes of death worldwide and sudden cardiac death (SCD) accounts for ~25% of these deaths. Despite its epidemiologic relevance, there are very few diagnostic strategies available useful to prevent SCD mainly focused on patients already affected by specific cardiovascular diseases. Unfortunately, most of these parameters exhibit poor positive predictive accuracy. Moreover, there is also a need to identify parameters to stratify the risk of SCD among otherwise healthy subjects. This review aims to provide an update on the most relevant non-invasive diagnostic features to identify patients at higher risk of developing malignant ventricular arrhythmias and SCD.

## 1. Introduction

While deaths associated with cardiovascular diseases have decreased over the past several decades, these disorders remain among the leading causes of death worldwide. Sudden cardiac death (SCD) accounts for ~25% of these deaths [1,2]. Likewise, and despite its epidemiologic relevance, there are very few diagnostic strategies available that can be used to prevent SCD. Numerous parameters have been evaluated to identify those at higher risk of SCD, with a particular focus on patients already diagnosed with specific cardiovascular diseases. Unfortunately, most of these parameters exhibit poor positive predictive accuracy. Moreover, there is also a need to identify parameters to stratify the risk of SCD among otherwise healthy subjects. For these reasons, there is a critical need for new parameters that can be used to identify individuals at high risk for SCD, particularly among those who might benefit from appropriate therapeutic strategies. In this clinical setting, it should be considered the information carried by the different available diagnostic tools. A multiparametric approach, based on the use of imaging and non-imaging techniques and able to more appropriately indentify those patients in whom a cardioverter defibrillator (ICD) implantation could exert the greatest beneficial effects (Figure 1).

This review aims to provide an update on the most relevant diagnostic features revealed by electrocardiography (ECG), imaging modalities, and genetic testing to identify patients at higher risk of developing malignant ventricular arrhythmias and SCD.

## 2. Electrocardiographic Features Associated with the Risk of Arrhythmias

ECG is an ideal tool for both diagnosis and prognostic stratification because it is non-invasive, cost-effective, and widely accessible. Currently, electrocardiographic findings used to prevent SCD are primarily those used to diagnose inherited channelopathies, such as long QT (LQTS) or Brugada (BrS) syndrome [3]. Over the last few years, several new electrocardiographic parameters have been identified that are associated with an increased risk of SCD mainly in patients with arrhythmogenic cardiomyopathies (Table 1).

### 2.1. QRS Fragmentation

While QRS duration is consistently associated with an increase in mortality of all causes [4,5], fragmented QRS complexes (fQRS) may specifically represent conduction disarray secondary to ventricular myocardial fibrosis and/or scarring [6]. fQRS was initially defined as additional spikes in the QRS complex in the absence of bundle branch block [7]. The definition was subsequently broadened to include QRS complexes with notches in addition to those detected in the pre-existing pattern. These notches can be observed even in wide QRS complexes resulting from bundle branch block, paced rhythm, or ventricular ectopy. In the latter case, this pattern is described as a fragmented wide QRS (f-wQRS), in contrast with the aforementioned fragmented QRS complexes of normal duration (i.e., <120 ms). An fQRS detected in a paced rhythm is known as f-pQRS [8]. A recent large meta-analysis that included 45 studies with a total enrollment of 6088 patients revealed that fragmentation of the QRS complex was independently associated with arrhythmic events, including SCD and malignant ventricular arrhythmias (odds ratio [OR], 6.73; 95% confidence interval [CI], 3.85–11.76; *p* <0.001) [9].

### 2.2. Early Repolarization (ER)

ER can be detected on an electrocardiogram by its characteristic J-point elevation of at least 1 mm in two contiguous leads. This finding can include a “notching” type appearance, i.e., a positive J-deflection inscribed within the S wave, or a “slurring”, i.e., a gradual rather than a distinct transition from the QRS to the ST segment in the inferior, lateral, or inferolateral leads [10,11]. ER was historically considered to be a benign finding. ER was observed frequently in the anterolateral leads of ECG tracings from young black male athletes and adolescents and could be accentuated by vagal tone and hypothermia [12,13,14]. However, recent studies have revealed that ER can be an inherited finding. Results from a study of 500 families revealed that subjects with at least one parent exhibiting an ER pattern on ECG were twice as likely to display the same variant. Family transmission is observed more frequently in subjects whose mothers have an ER pattern. Of importance, these families exhibit a comparatively high incidence of sudden death which may be linked to ER [15].

Malignant ER may predispose affected individuals to arrhythmias and fatal events due to re-entry phenomena, premature beats, and/or transmural dispersion [16]. Thus, it is important to recognize ECG characteristics that differentiate a relatively benign ER pattern from one that is malignant. Benign forms of ER include rapidly ascending ST-segments that blend with T-wave associated with a normal QT interval (QTc). By contrast, ECG tracings in which ST segments remain flat, horizontal, or descend towards the T-wave with a prolonged QTc are diagnostic of the more malignant variant of ER that has been associated with mortality due to arrhythmias in long-term follow-up [17,18]. Likewise, ER with a deflection in the R-wave (slurred pattern) descending in the terminal part of the QRS observed in tracings from at least two inferior leads (II, III, aVF) and two lateral leads (I, aVL, V4–V6) has been associated with an increased risk of idiopathic ventricular fibrillation (IVF) [19,20]. This pattern has been identified as an independent predictor of fatal arrhythmic events in patients diagnosed with Brugada syndrome (BrS) as described below [21,22].

### 2.3. Brugada Syndrome (BrS)

ECG plays a key role in the diagnosis of the rare but potentially lethal condition known as Brugada Syndrome (BrS). The diagnosis of BrS is based on the detection of one of two types of aberrant ECG patterns. A Brugada type 1 ECG pattern exhibits a prominent coved ST-segment elevation displaying a J-point amplitude or ST-elevation ≥2 mm in more than one of the right precordial leads (V1–V3) followed by a negative symmetric T-wave. Patients with spontaneous type 1 ECG patterns are at higher risk of arrhythmic events than those displaying the drug-induced pattern [23,24]. The risk of arrhythmias in these individuals ranges from 0.81%/year in those who remain asymptomatic to 2.3%/year in symptomatic patients [25].

Recently several ECG parameters have been proposed to identify individuals at higher risk for developing these negative sequelae. QRS fragmentation (f-QRS) in precordial leads V1 and/or V2–V3 has been recognized as an independent predictor for future arrhythmias in patients diagnosed with BrS [26,27]. QRS prolongation in leads II, V2, and V6 is also associated with an increased risk of developing arrhythmic events [28].

The “aVR sign” has also been identified as a potential marker of arrhythmic risk in this syndrome. The aVR sign features an R wave ≥0.3 mV or R/q ≥ 0.75 in the aVR lead. Mechanistically, this finding might reflect a right ventricular conduction delay associated with an increased risk of arrhythmic events in BrS patients [29]. The specific relevance of the aVR lead in these cases relates to its focus on events in the right ventricular outflow tract (RVOT); electroanatomical abnormalities at this site may predict the inducibility of ventricular fibrillation in patients diagnosed with BrS [30]. Lateral leads, including L1 and aVL, provide complementary information that focuses on depolarization and repolarization of the RVOT. Similarly, the presence of an S wave in lead I ≥40 ms with amplitude ≥0.1 mV and area ≥1 mm^2^ in patients with BrS strongly predicted both VF and SCD with good sensitivity albeit with low specificity [31].

Finally, a prolonged QTc, a longer interval between the peak and the end of the T-wave (Tpeak-Tend interval), a higher Tpeak-Tend/QT ratio, and a higher Tpeak-Tend dispersion in precordial leads [31,32,33,34], first-degree atrioventricular block (AVB) [35], and spontaneous atrial fibrillation (AF) [36] have all been associated with an increased risk of significant arrhythmic events.

### 2.4. T-Wave Morphology in Long QT Syndrome (LQTS)

LQTS is typically diagnosed on repeated 12-lead ECGs that exhibit either a QTc ≥480 ms or a calculated LQTS risk score >3 in patients with a confirmed pathogenic LQTS mutation [37]. Electrocardiographic parameters have been studied in patients diagnosed with this disorder to stratify arrhythmic risk. Among these parameters, T-wave morphology may be a relevant feature with respect to the increased arrhythmic risk exhibited among patients affected by one or more of these inherited channelopathies. For example, female carriers of type 2 LQTS with abnormal T-waves had a significantly higher risk of cardiac events compared to those with normal T-wave morphology; interestingly, this association was not significant in males [38]. In these cases, T-wave morphology was evaluated in leads V5 and II and classified as normal, broad, flat, notched, or biphasic.

### 2.5. Electrocardiographic Markers in Arrhythmogenic Right Ventricular Dysplasia/Cardiomyopathy (ARVD/C)

A recent systematic review and meta-analysis considered the current status of electrocardiographic markers to assess risk in patients diagnosed with ARVD/C [39]. Among the conclusions reached, the presence of non-sustained VT (NSVT) and the extent of T-wave inversion (TWI) on anterior and/or inferior leads can predict the likelihood of the development of sustained ventricular arrhythmias (VAs) and SCD [40,41]. While epsilon waves may reflect the short-term risk of developing critical arrhythmias, the use of this marker is limited by its low sensitivity and specificity, and its dependence on ECG filter setting and magnification [42].

Terminal activation duration (TAD) is measured from the nadir of the S wave to the end of all depolarization deflections; prolonged TAD is defined as a duration of ≥55 ms in any of the V1–V3 leads in the absence of complete right bundle branch block. Of note, TAD prolongation was the only ECG abnormality detected in four family members (of seven) who carried a genetic variant associated with ARVD/C. These findings suggest that TAD may be used as a means for early recognition of at-risk individuals [43].

## 3. Echocardiographic Markers for Arrhythmias

Echocardiography is the first and the most commonly used cardiac imaging technique. Compared to cardiac magnetic resonance and cardiac computed tomography, echocardiography is an inexpensive, rapid, and readily available imaging modality. Echocardiographic imaging of patients with ventricular arrhythmias facilitates the identification (or exclusion) of structural heart disease. Furthermore, echocardiography performed in patients who are exercising or responding to pharmacological stress can be applied to a selected group of patients with VAs triggered by ischemia [1,37].

Currently, left ventricular ejection fraction (LVEF) is the main echocardiographic parameter used to stratify the risk of SCD. Current guidelines recommend an implantable cardioverterdefibrillator (ICD) for patients who present with a low LVEF [44]. However, SCD frequently occurs in the absence of reduced LVEF. In the community-based Oregon Sudden Unexpected Death Study, almost half of all cases of SCD (48%) were individuals who had normal LVEFs (55%), with less than a third exhibiting severely reduced LV systolic function [45]. Furthermore, results from several studies revealed that less than a third of the recipients with LVEF ≤35% received effective therapeutic benefit from an ICD, while subject to the well-known complications of this device [46].

Evaluation of additional risk factors revealed by echocardiography might help to identify patients at higher risk of SCD. Considering the widespread availability of this imaging modality, a larger understanding of these potential risk factors could have a substantial impact on the health and well-being of the general population [47]. Newly identified echocardiographic parameters have been proposed that may offer a more accurate evaluation of ventricular function and a more effective means to prevent SCD (Table 2). However, the clinical utility of these markers needs to be examined in randomized clinical trials.

### 3.1. Left Ventricular Hypertrophy (LVH)

Increased left ventricular (LV) mass is associated with an increased risk of SCD in the general population [48,49]. LVH is associated with abnormal interstitial remodeling which can impair electrical conduction and promote Vas [50]. In the Framingham Heart Study, increased LV mass and hypertrophy as assessed by echocardiogram, were independently associated with SCD after accounting for other known risk factors [51]. LVH, defined as LV mass (adjusted for height) >143 g/m^2^ in men and >102 g/m^2^ in women, was associated with a 116% increase in the risk of SCD. Estimates revealed that each 50 g/m^2^ incremental increase in LV mass was associated with an additional 45% increase in the risk of SCD during a mean follow-up of 10.3 years. The increased risk of SCD associated with LVH is also observed among patients diagnosed with ischemic cardiomyopathy [52].

### 3.2. Global and Segmental Longitudinal Strain

Several new echocardiographic parameters reflecting myocardial function have been developed in recent years. Among these are speckle tracking echocardiography (STE) which provides information on myocardial deformation by linking electrical activity to specific anatomical structures. Active myocardial deformation (strain) is assessed by tracking ultrasound markers (speckles). The analysis focuses on LV longitudinal strain and presents a value for global longitudinal strain (GLS). GLS has been independently associated with SCD, appropriate ICD therapy, and VA in patients diagnosed with ischemic cardiomyopathy, cardiac systemic sclerosis, and in patients who have undergone repair of tetralogy of Fallot. Detection of impaired segmental longitudinal strain in the peri-infarct zone is also independently associated with an increased risk of the need for ICD therapy to treat either VF or VT [53].

The transitional zone between necrotic and healthy tissue may play an important role in the pathophysiology of VAs and SCD. Mixed scar and viable tissue in peri-infarct areas represent a potential substrate for electrical re-entry and lower values of longitudinal strain on STE. Thus, detection of lower levels of longitudinal strain specifically in peri-infarct zones will help to identify subjects at increased risk for VF and VT [53,54]. Overall, reductions in parameters that reflect regional longitudinal myocardial deformation may provide incremental prognostic information beyond that provided by clinical and conventional echocardiographic risk factors regardless of whether the origins of the cardiomyopathy were ischemic or non-ischemic [55,56].

### 3.3. Mechanical Dispersion

STE can also be used to measure the time to peak of longitudinal strain and a calculation of LV mechanical dispersion (LVMD). Mechanical dispersion reflects the heterogeneity of myocardial contraction as a product of electrical alterations and tissue abnormalities. High levels of LV mechanical dispersion suggest the presence of slow and heterogeneous electrical conduction of the LV myocardium (e.g., secondary to scar tissue). After a myocardial infarction, the combination of LV GLS and LVMD may suggest the need for an ICD, even in patients with an LVEF >35% [57]. Moreover, an LVMD ≥ 75 ms may be an independent predictor of SCD and malignant VAs regardless of an ischemic etiology, even in patients with LVEF > 35% [58]. Measurements of LVMD and LV GLS of hypertrophic segments may prove to be a useful tool to also stratify the risk of SCD among patients diagnosed with hypertrophic cardiomyopathy [59,60] and mitral valve prolapse [61]. Finally, STE can also be used to identify abnormalities of right ventricular (RV) function. Both RV GLS and RVMD are associated with malignant Vas [62,63].

## 4. Cardiac Magnetic Resonance (CMR)

CMR provides the most complete evaluation of a patient’s cardiac status with good temporal and spatial resolution that facilitates the quantitative assessment of chamber size, myocardial wall thicknesses, ventricular function and mass, and segmental function, as well as the identification of anomalous coronary arteries. CMR can be used to diagnose myocarditis, amyloidosis, sarcoidosis, Chagas disease, Fabry disease, LV non-compaction cardiomyopathy (LVNC), hemochromatosis, and arrhythmogenic cardiomyopathy.

Numerous studies have considered the relevance of CMR for the study of arrhythmogenic factors that might predict SCD (Table 3, Figure 2). In particular, techniques including late gadolinium enhancement (LGE), T1 mapping, and T2 mapping, as well as those that evaluate extracellular volume can be used to identify structural changes, storage, infiltration, inflammation, fibrosis, and scarring.

### 4.1. Late Gadolinium Enhancement

LGE detected 15 min after collection of contrast-enhanced CMR sequences can be used to identify patients with both ischemic and non-ischemic dilated cardiomyopathy who are at increased risk of developing malignant Vas [64,65,66,67]. The association between LGE and an arrhythmic endpoint was presented in studies in patients with LVEFs < 35% but may also be applicable in patients with a mean LVEF >35% [68]. LGE detected in patients diagnosed with myocarditis suggests an increased risk of SCD regardless of LVEF [69,70,71].

Various LGE features are relevant for arrhythmic risk stratification (e.g., findings documenting its extension, localization, and size of the border zone). While risk is increased under conditions in which LGE exceeds to include more than 5% of LV mass, detection of LGE may be relevant even at lower percentages [72,73]. There are only a few studies that have examined the association of total LGE with arrhythmic risk in patients diagnosed with chronic myocarditis. In these studies, the size of the scar was directly associated with the probability of manifesting VAs [74]. Anterior and septal as well as mid-wall LGE locations have been associated with arrhythmic events in patients with non-ischemic cardiomyopathy [75,76]. The area surrounding the region of LGE, known as the “border zone”, consists of viable and nonviable myocytes separated by scar/fibrotic tissue involved in the development of arrhythmias [77,78]. The characteristics of the border zone can predict the inducibility of VT on electrophysiological studies (EPSs), the need for appropriate ICD therapy, and the likelihood of SCD, all independently of the LVEF [79,80,81,82,83]. LGE could also be useful in patients diagnosed with hypertrophic cardiomyopathy (HCM); replacement fibrosis that is commonly detected in these patients has been associated with an adverse prognosis [84,85].

CMR plays an important role in the differential diagnosis of HCM versus athlete’s heart. This modality can be used to measure LV wall thickness and detect LV hypertrophy that was not detected on echocardiography. A diagnosis of athlete’s heart can be made in patients who decondition over time in association with regression in cardiac wall thickness >2 mm. By contrast, a diagnosis of HCM is suggested in cases in which the hypertrophy remains unchanged. Likewise, LGE and a focal area of LV hypertrophy that remains unchanged following deconditioning also support a diagnosis of HCM. LV remodeling associated with a diagnosis of athlete’s heart typically does not result in focal areas of myocardial scarring, especially in younger individuals. However, although the detection of LGE on contrast-enhanced CMR favors the diagnosis of HCM, the absence of LGE cannot be used to exclude HCM, as this finding has been reported in only half of patients with this clinical diagnosis.

LGE assessments are limited by their semi-quantitative nature and the fact that they can provide only an estimate of irreversible myocardial damage. The methods used to detect and report LGE have not been standardized. Furthermore, LGE may be a dynamic parameter. For example, the expansion of the extracellular volume at early timepoints after cardiac injury leads to an increased volume of gadolinium distribution; as such, this finding reflects not only the fibrosis at this stage but also the interstitium. Finally, CMR is not widely available and impaired renal function is a relative contraindication for the administration of gadolinium-based contrast agents.

### 4.2. Mapping Techniques and Extracellular Volume

T1 and T2 mapping methods provide a quantitative approach to the assessment of cardiac tissue and reflect the magnetic properties of cardiac muscle based on its composition. T1 values are increased by tissue edema and fibrosis and are reduced by lipid (e.g., Anderson-Fabry disease) and iron overload. T1 mapping also detects diffuse fibrosis associated with both ischemic cardiomyopathy and NICM and has been explored as an independent predictor of sustained VT and the need for appropriate ICD therapy [86]. Unlike LGE, native T1 values are frequently abnormal in diffuse diseases of the myocardium and thus can provide insights into the etiology and pathogenesis of NICM. A T1 map may highlight focal areas of edema such as those accompanying acute myocarditis, acute myocardial infarction, or Takotsubo cardiomyopathy. LGE can be used to evaluate ECV, even if the role of this value in determining the risk of SCD risk remains to be evaluated. Postcontrast T1 mapping techniques combined with measurements of native T1 and hematocrit provide an estimation of the ECV [87]. Myocardial fibrosis identified by ECV measurements may be associated with hospitalization and death secondary to heart failure (HF) [88]. The ECV is also elevated in regions of chronic infarction. The main advantage of T1 mapping over LGE for the stratification of the risk of arrhythmias is the possibility of using this modality to identify diffuse myocardial fibrosis in the setting of NICM [89]. The evaluation of pre-contrast (native) and post-contrast T1 mapping images can identify diffuse myocardial fibrosis that remains undetectable by LGE [90,91].

T2-weighted sequences identify and provide a quantitative assessment of myocardial edema. There are several technical limitations to this method, including its non-standardized assessment. However, initial data suggest that an abnormal T2 mapping can be used to predict major adverse events including SCD and the need for cardiac transplantation and/or implantation of a ventricular assist device.

### 4.3. Feature Tracking (FT)

FT is a recently developed postprocessing tool that can be applied to CMR to acquire information on strain parameters regardless of contrast agent. One recent study revealed that measurements of LV GLS and RV-global radial strain can be used to predict outcomes in patients diagnosed with ischemic cardiomyopathy with an LVEF > 35%. This information might assist with decision making regarding ICD use in patients with ischemic cardiomyopathy with a mild or moderately reduced EF [92].

### 4.4. CMR in Mitral Valve Prolapse

Mitral valve prolapse (MVP) is rarely associated with SCD. In addition to the aforementioned echocardiographic parameters, CMR may also elucidate features that reflect an increased risk of malignant arrhythmias. Myocardial fibrosis associated with arrhythmias as detected by LGE [93] is mainly found at the LV posteromedial papillary muscle or the level of the inferobasal wall [94,95]. Subclinical diffuse ventricular fibrosis (which may be a precursor of focal fibrosis in MVP or a different disease in which fibrotic markers undergo up-regulation) may be a marker for early identification of patients at risk of SCD [96]. Disjunction of the mitral annulus (i.e., detachment of the root of the annulus from the ventricular myocardium located at the base of the posterior leaflet) is a pro-arrhythmogenic event [97]. Finally, LGE findings may be used to predict future adverse cardiac events associated with this finding in other clinical settings such as sarcoidosis [98,99,100,101,102]. 

## 5. Genetic Testing

The number of genetic defects that have been associated with the pathogenesis of inherited cardiomyopathy has been increasing. Moreover, the ongoing development of new techniques for rapid evaluation of the human genome suggests that genetic analysis will be even more relevant in future clinical practice. Recent advances in DNA sequencing technologies have made it possible to explore large numbers of disease genes simultaneously as mutational analysis proceeds much more rapidly and at a reduced cost. These new methods, collectively known as next-generation sequencing (NGS), represent a major advance in our ability to identify causative mutations in families affected by genetic disorders [103,104]. The second relevant aspect concerning the progress of genetic testing to prevent SCD is the association between specific genetic polymorphisms and aberrancies with malignant VAs. As shown in Table 4, genetic mutations have been identified that are associated with an increased risk of VAs in patients diagnosed with cardiomyopathy [105]. More than one hundred specific genes identified in patients with dilated cardiomyopathy (DCM) have been evaluated [106,107,108]; some of these were strongly associated with both DCM and specific electrical phenotypes [109]. Mutations in the gene encoding the nuclear structural protein lamin a/c (LMNA) have been identified as among the leading causes of an increased risk of arrhythmias in patients diagnosed with DCM [110]. Mutations in LMNA have also been implicated in several distinct phenotypes of familial cardiomyopathy and a high incidence (46%) of SCD [111,112]. Based on these data, both European and American guidelines suggest that ICDs might be provided to patients with LMNA mutations with other risk factors, including non-sustained VT, LVEF < 45%, non-missense mutations, and male sex, even in the absence of severe LV dysfunction. DCM patients with variants in the gene encoding the critical muscle protein, titin (TTN) were also at an increased risk of Vas [113,114]. Likewise, mutations in the gene encoding the sodium voltage-gated channel alpha subunit 5 (SCN5A) correlated with the development of supraventricular arrhythmias (86%), sick sinus syndrome (33%), AF (60%), VT (33%), and conduction disease (60%) also in patients diagnosed with DCM [115,116]. Analogously, patients with DCM and mutations in the gene encoding filamin C (FLNC) also have a notably high incidence of VA and SCD [117,118].

Genetic variants have also been identified to beassociated with malignant arrhythmias in other cardiomyopathies. For example, mutations in numerous genes encoding sarcomere proteins have been identified in patients diagnosed with HCM [119]. Among these is the Val606Met mutation in the cardiac beta-myosin heavy chain that was identified in a patient cohort diagnosed with familial HCM. This specific mutation has been associated with a high risk of SCD at a young age [120]. Similarly, mutation in the gene encoding cardiac myosin-binding protein C (MyBP-C) has been reported in patients with HCM and substantial hypertrophy and has been associated with a moderate incidence of SCD; by contrast, the Arg820Gly mutation has been associated with end-stage HCM [121].

Available data suggest that patients diagnosed with ARVC with multiple mutations in desmosomal genes are likely to have a more severe phenotype and an increased lifetime risk of malignant arrhythmias and SCD; interestingly, healthy carriers of this gene are considered to be at low risk for this outcome. The current International Task Force recommendations stratify patients with ARVC into high-, intermediate-, and low-risk groups as a basis for guiding decisions regarding ICD implantation [122,123,124]. In the case of ARVD, pathogenic mutations represent one variable of many that have been linked to the more severe progression of the disease; mutational data can be included together with several other well-defined variables to create an appropriate risk stratification model and to improve the evaluation of the overall risk of SCD [125].

Subunit alpha of voltage-gated sodium channel 5A plays an important role in cardiac myocyte depolarization. Pathologic mutations in this gene were first reported in 1995 and were originally associated with electrical cardiac pathologies, including LQTS [126], BrS [127], and Lenegre’s disease [128]. This mutation has also been reported in small cohorts of patients diagnosed with DCM [129], AVRC [130], and LVNC [131]. All patients in these cohorts have a higher risk of both supraventricular arrhythmias (SVAs) and VAs; consideration for ICDs has already been recommended although this issue is not addressed in the current guidelines [132]. Finally, mutations in CPVT1 which encode the cardiac ryanodine receptor (RyR2) have been identified in patients with autosomal dominant for catecholaminergic polymorphic ventricular tachycardia (CPVT). The autosomal recessive form, which is less common, results from mutations in the gene encoding cardiac calsequestrin 2 (CASQ2) [133,134]. It will be critical to assess the role of these mutations as there are very few risk factors available to provide prognostic information in patients diagnosed with CPVT [135,136].

Targeted genetic analysis of ion channels (i.e., RYR2, KCNQ1, KCNH2, and SCN5A) should be considered to establish the probable cause of death in patients with SCD of unclear etiology as well as to facilitate the identification of relatives who are potentially at risk [137]. With this information, family screening of first-degree relatives could represent an effective strategy to prevent SCD [138]. Targeted molecular screening in first-degree relatives might also be considered when there is the suspicion of the presence of an inheritable disease in family members. On the other hand, genetic screening of large panels of genes should not be performed without clinical clues for a specific disease or pathology. In most cases, only a fraction of potentially affected family members undergo screening, in part due to the lack of appropriate infrastructure but also secondary to the levels of anxiety and distress that may result from a positive finding [139]. Decisions regarding the need to proceed with genetic testing should be based on the clinical value that genetic information might provide to the care of individual patients and their families. In most cases, genetic testing is not necessary to establish a diagnosis. The main point of genetic testing in this field currently is to facilitate screening of family members who might carry hereditary heart diseases and to identify cases of subclinical disease that might require medical surveillance. In some circumstances, genetic tests can help establish a diagnosis in cases of equivocal clinical presentations. However, it is critical to bear in mind that genetic information requires careful interpretation and is frequently inconclusive. Considering the low prevalence of hereditary arrhythmic syndromes, clinical evaluation and decisions relating to the usefulness of genetic tests should be carried out by physicians with dedicated and specific skills [140]. The term “personalized medicine” was recently introduced to refer to the integration of clinical, molecular, and environmental markers associated with the risk of developing a specific disease. At present, clinical genetics can play a role in defining a personalized approach to the prevention of SCD via the integration of clinical, environmental, and molecular data [141].

## 6. Conclusions

Cardiac disease can lead to SCD via various etiologies that converge on a final pathway that ultimately leads to fatal VAs. While there are no models for the prediction of SCD risk among those in the general population, there are numerous studies that describe individual risk factors for patients diagnosed with cardiac disease. Identification and treatment of these underlying conditions and determining how to optimize cardiac function remain essential parameters in the ongoing effort to prevent SCD. Secondary and primary prevention are best achieved by ICD implantation. In this setting, there is the clinical need to identify patients in whom ICD implantation may be more beneficial. On the basis of a systematic review of the literature, this review summarized theparameters derived from different non-invasive techniques which can be used to support risk-stratification strategies in patients. The goal was to provide an update useful in the routine clinical practice, which focused on the parameters derived from the more easily accessible tests, such as ECG and echocardiography, to more advanced ones, such as CMR and genetic testing.

The goal for the future is to move forward from population-based management to personalized therapeutic strategies. To this end, risk stratification algorithms based on findings from epidemiological studies that evaluate traditional cardiovascular risk factors, lifestyle, and dietary habits, as well as imaging, biological markers, and genetic variants alone or in combination may aid in the identification of susceptible subgroups within a given population. Improvements in epidemiological studies, experimental investigations, and clinical trials will be essential to achieve strategies aimed at reducing the incidence and lethality of SCD across the entire population.

## Figures and Tables

**Figure 1 jcm-11-01519-f001:**
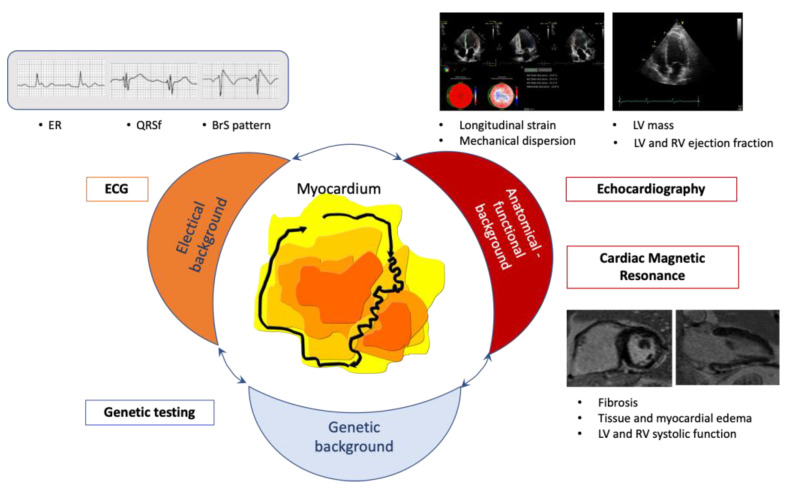
Figure summarizes the features related to increased risk of ventricular arrhythmias obtained by ECG, imaging (echocardiography and cardiac magnetic resonance) and genetic techniques. BrS: Brugada Syndrome; ER: early repolarization; LV: left ventricular; QRSf: fragmented QRS.

**Figure 2 jcm-11-01519-f002:**
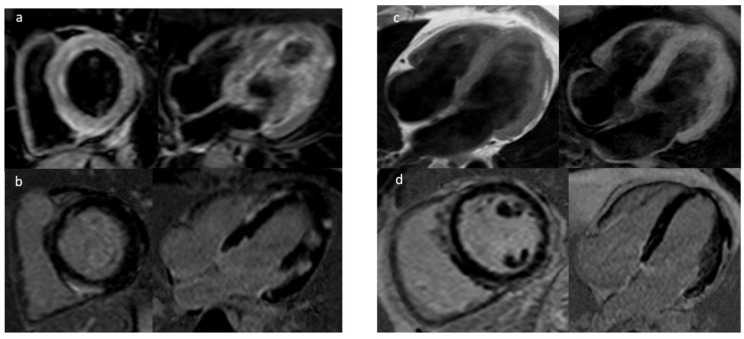
In the left panel T2-STIR (**a**) and PSIR-TFE (**b**) sequences showing increased signal predominantly subepicardial patchy of the left ventricular wall due to necrosis of acute myocarditis. In the right panel sequence T1-TSE and T1-Fat Sat (**c**) showing multiple areas of adipose infiltration with mesocardial and subepicardial distribution of the left ventricle walls, (**d**) extended subepicardial signal hyperintensity in PSIR sequences for the study of “Late Gadolinium Enhancement” indicative of fibrosis for Left Dominant Arrhythmogenic Dysplasia.

**Table 1 jcm-11-01519-t001:** ECG findings associated with an increased arrhythmic risk.

ECG Abnormality	Pathophysiologic Background	Clinical Setting
QRSf	Conduction delay from inhomogeneous activation of the ventricles due to myocardial scar	DCM, IDCM HCM, BrS LQTS, ARVD Cardiac Sarcoidosis
ER (“J-waves” or “J-point elevation”)	Altered ion channel function (alterations in sodium, potassium and calcium currents)	Young African men Athletes HCM
TW Inversion	Changes during phase three of the action potential	ARVD

BrS: Brugada syndrome; ER: early repolarization; ICM: ischemic cardiomyopathy; NICM: non-ischemic cardiomyopathy; HCM: Hypertrophic cardiomyopathy; ARVC: arrhythmogenic right ventricular cardiomyopathy; QRSf: QRS fragmentation; TW: T wave.

**Table 2 jcm-11-01519-t002:** Echocardiographic parameters associated with an increased risk of malignant ventricular arrhythmias.

Echocardiography	Pathophysiologic Background	Clinical Setting	Information
LVEF	Left ventricular systolic function	ICM NICM Myocarditis ARVC LVNC	Severe LV systolic dysfunction, of any cause, identified by measuring the LVEF, is associated with an increased risk of SCD (LVEF < 35%)
LV GLS/RLS	Measure of LV systolic function (indirect reflector of myocardial fibrosis/scar)	ICM NICM HCM	GLS is associated with SCD, appropriate ICD therapy and VA
Mechanical dispersion	Slow and heterogeneous electrical conduction of the LV myocardium (indirect reflector of myocardial fibrosis/scar)	ICM NICM HCM ARVC	Predictor of VA in patients with moderate and severe LV systolic dysfunction (despite the etiology of LV dysfunction) and in HCM patients. Predictor of VT/VF (in patients with ARVC)
LV wall thickness	Left ventricular hypertrophy	HCM Myocarditis	Independent predictor of SCD
RVEF, RV diameter, regional RV akinesia, dyskinesia or aneurism	RV remodeling and dysfunction	ARVC	Correlated with more frequent sustained ventricular arrhythmias and ICD appropriate shocks

CMR: cardiac magnetic resonance; LV: left ventricular; GLS/RLS: left ventricular global longitudinal strain/regional longitudinal strain; ICD: implantable cardioverter defibrillator; ICM: ischemic cardiomyopathy; NICM: non-ischemic cardiomyopathy; HCM: Hypertrophic cardiomyopathy; ARVC: arrhythmogenic right ventricular cardiomyopathy; LVNC: left ventricular noncompaction; RV: right ventricular; RVEF: right ventricular ejection fraction; SCD: sudden cardiac death.

**Table 3 jcm-11-01519-t003:** Imaging parameters associated with an increased risk of malignant arrhythmias.

CMR	Pathophysiologic Background	Clinical Setting	Information
LGE	Fibrosis	ICM, NICM HCM, Myocarditis ARVC, LVNC Mitral valve prolapse	Independent predictor for VA and SCD
T1 and ECV	Tissue edema and diffuse fibrosis	ICM, NICM HCM, Myocarditis	Higher native T1 values associated with VA
T2	Myocardial edema	Myocarditis	Abnormal T2 mapping is involved in predicting major adverse events including cardiac death
LVEF	Left ventricular systolic function	ICM, NICM Myocarditis, ARVC LVNC	LV systolic dysfunction is associated with an increased risk of SCD
RVEF	Right ventricular systolic function	ARVDC	Overall increase in VA in RV dysfunction
Strain Imaging and MD	Myocardial deformation and function	ICM, NICM	Impaired strain associated with SCD

CMR: cardiac magnetic resonance; LGE: late Gadolinium enhancement; ICM: ischemic cardiomyopathy; NICM: non-ischemic cardiomyopathy; HCM: hypertrophic cardiomyopathy; ARVC: arrhythmogenic right ventricular cardiomyopathy; LVNC: left ventricular noncompaction; RVEF: right ventricular ejection fraction.

**Table 4 jcm-11-01519-t004:** Genes associated with an increased risk of malignant arrhythmias and SCD in inherited cardiomyopathy.

Inherited Cardiomyopathy	Genes Associated with SCD	Protein Encoded
DCM	TTN	Titin
	LMNA	Lamin A/C
	FLNC	Filamin C
	SCN5A	Sodium voltage-gated channel alpha subunit 5
HCM	MYH7	B-Myosin Heavy Chain 7
	MYBPC3	Myosin-Binding Protein C 3
BrS	SCN5A	Nav1.5—Cardiac sodium channel α subunit
ARVD	PLN R14del	Phospholamban
	LMNA	Lamin A/C
	SCN5A	Sodium voltage-gated channel alpha subunit 5
	RBM20	RNA binding motif protein 20
	TMEM43	Transmembrane Protein 43
LQTS	SCN5A	Nav1.5—Cardiac sodium channel α subunit

SCD: sudden cardiac death; BrS: Brugada Syndrome; DCM: dilated cardiomyopathy; HCM: hypertrophic cardiomyopathy; LQTS: long QT syndrome; ARVD: arrhythmogenic ventricular dysplasia.

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
