# Peer review of "Recent Non-Invasive Parameters to Identify Subjects at High Risk of Sudden Cardiac Death"

_jcm, 2022, doi:10.3390/jcm11061519_

Round 1

Reviewer 1 Report

  • It's a well structured and nicely written article. However, conclusion part is not that clear.
  • If authors want to figure out more accurate  parameters/features for positive prediction of SCD, they do not  mention why they picked up only Electrocardiographic and genetic testing as features only for  SCD  prediction. 
  • They also do not mention, any clear message of this review. How this review is better then similar articles published earlier

Author Response

We would like to thank the reviewer for her/his helpful comments. This is our point to point reply:
- It's a well structured and nicely written article. However, conclusion part is not that clear.
We thank the reviewer for having appreciated our work. We modified the conclusions to make them clear.   - If authors want to figure out more accurate  parameters/features for positive prediction of SCD, they do not  mention why they picked up only Electrocardiographic and genetic testing as features only for  SCD  prediction. In the review beside ECG and genetic testing, we have also focused on the parameters coming from imaging techniques, i.e. echocardiography and magnetic resonance. We highlighted it in introduction and in conclusions.
- They also do not mention, any clear message of this review. How this review is better then similar articles published earlier
In the conclusions we tried to stress the original aspects of our review.

Reviewer 2 Report

I feel very honoured to have the opportunity to review the manuscript entitled “Recent non-invasive parameters to identify subjects at high risk of sudden cardiac death.” The authors had systematically elaborated the predictive value of non-invasive diagnostic features such as ECG parameters (fragmented QRS, early repolarization, Brugada-like ECG, T wave morphology), echocardiographic measurements (left ventricular hypertrophy, global longitudinal strain, mechanical dispersion), CMR imaging modalities (LGE, T1 / T2 mapping, strain imaging) and genetic testing, to estimate the risk of SCD. This manuscript was well written and may help increase the understanding of SCD and promote the clinical application of these novel diagnostic tools such as LGE imaging to estimate the risk of SCD for general patients or specific subgroups. Moreover, the Figures and Tables were vividly presented, providing a variety of valuable information. 

Author Response

I feel very honoured to have the opportunity to review the manuscript entitled “Recent non-invasive parameters to identify subjects at high risk of sudden cardiac death.” The authors had systematically elaborated the predictive value of non-invasive diagnostic features such as ECG parameters (fragmented QRS, early repolarization, Brugada-like ECG, T wave morphology), echocardiographic measurements (left ventricular hypertrophy, global longitudinal strain, mechanical dispersion), CMR imaging modalities (LGE, T1 / T2 mapping, strain imaging) and genetic testing, to estimate the risk of SCD. This manuscript was well written and may help increase the understanding of SCD and promote the clinical application of these novel diagnostic tools such as LGE imaging to estimate the risk of SCD for general patients or specific subgroups. Moreover, the Figures and Tables were vividly presented, providing a variety of valuable information.
We would like to thank the reviewer for having appreciated our work, we are very honored from this.